# An Overview of Postoperative Intraabdominal Adhesions and Their Role on Female Infertility: A Narrative Review

**DOI:** 10.3390/jcm12062263

**Published:** 2023-03-15

**Authors:** Stefan Ghobrial, Johannes Ott, John Preston Parry

**Affiliations:** 1Clinical Division of Gynecologic Endocrinology and Reproductive Medicine, Medical University of Vienna, 1090 Vienna, Austria; 2Parryscope and Positive Steps Fertility, Madison, Madison, WI 39110, USA; 3Department of Obstetrics and Gynecology, University of Mississippi Medical Center, Jackson, MS 39216, USA

**Keywords:** postoperative adhesion formation, infertility, complications, prevention, adhesion barrier

## Abstract

Postoperative intraabdominal adhesions can occur after more than 90% of gynecologic surgeries. They not only cause chronic pelvic pain and small bowel obstruction, but are also one of the main reasons for infertility. Adhesions are not only a burden for the affected patients, but are also a burden for the healthcare system, since the treatment of adhesion-associated complications costs a considerable amount of money. The gold standard for the diagnosis of adhesions is by laparoscopy, although other methods, such as transvaginal hydro-laparoscopy, are being discussed as better alternatives. Ideally, adhesions are avoided inherently, by operating carefully and by using microsurgical principles. If this is not possible, gel barriers have been shown to be successful in reducing postoperative adhesions.

## 1. Introduction

According to the World Health Organization (WHO), infertility is defined as the inability to achieve pregnancy after 12 months of regular, unprotected sexual intercourse [1]. It is estimated that 50–80 million women worldwide suffer from infertility each year [2]. Intraabdominal or peritoneal (pelvic) adhesions are a common cause for subfertility, through potential tubal occlusion and ovarian encapsulation [3,4]. Adhesions are fibrous connections between different adjacent tissues and organs, which usually result from inflammatory causes, most commonly after infection or surgery, and can occur in any part of the body [4]. The incidence of adhesions ranges from 67–93% after general abdominal surgery and can have rates up to 97% after open gynecologic pelvic procedures [5].

Although postoperative adhesions are common, neither an official definition nor a standardized classification can be found in the literature. Accordingly, this hinders the precision, interpretation, and generalizability of studies. Though the negative influence of intraabdominal adhesions on female fertility is well known [2,3,4,5], clinical guidelines for their diagnosis, treatment and prevention are not available [3]. This review focuses on published data for intraabdominal/pelvic adhesions in gynecology, with an emphasis on female subfertility/infertility.

## 2. Materials and Methods

A computerized search of the published literature from the Medline database was conducted in March 2022. No date or language restrictions were initially used. The search strategy used combinations of search terms and medical subject headings (MeSH) that were related to diseases and surgical interventions common in gynecology, including intraabdominal adhesions and infertility. The following search terms were used to find all relevant articles on this topic:

((“gynecologic* surg*”[tw] OR “Gynecologic Surgical Procedures”[Mesh] OR myomectom*[tw] OR “ovarian cyst*”[tw] OR cystectom*[tw] OR ovariectom*[tw] OR salpingotom*[tw] OR salpingectom*[tw] OR hysterectom*[tw]) AND (“intraabdominal adhesion*”[tw] OR “intra-abdominal adhesion*”[tw] OR “peritoneal adhesion*”[tw] OR “pelvic adhesion*”[tw] OR (“Tissue Adhesions”[Mesh] NOT intrauterin* NOT Asherman*))) AND (infertility[tw] OR sterility[tw] OR “Infertility, Female”[Mesh]). The literature search retrieved 261 references for possible inclusion in this review. After excluding non-English articles, 211 remained. Of these, we identified and reviewed 67 articles that were relevant and/or addressed the primary research question. We also reviewed references from these articles for additional information.

## 3. Types of Adhesions

An initial distinction should be made between congenital and acquired adhesions. Congenital adhesions can occur during organogenesis or are due to abnormal embryonic development of the peritoneal cavity. They usually cause no symptoms and are only diagnosed incidentally [3]. Acquired adhesions can result from both postoperative and non-operative inflammatory processes. Non-surgical causes of adhesions include a variety of inflammatory sources, such as pelvic inflammatory disease, peritonitis, cholecystitis, and diverticulitis. However, endometriosis, infections, and even complications from intrauterine contraceptives can also cause an inflammatory response and lead to pelvic adhesions. The true attributable proportion for each of these is difficult to assess, but it is believed that the majority of adhesions occur post-surgically [5,6].

## 4. General Significance of Intraabdominal Adhesions

The extent of adhesion-related problems is best illustrated by a study conducted by the UK Surgical and Clinical Adhesions Research (SCAR) group, showing that 5.7% (1209) of all readmissions (21,347) of patients undergoing open abdominal or pelvic surgery were classified as being directly related to adhesions, with 3.8% (1169) of the patients being managed operatively. Moreover, 34.6% of the patients who underwent open abdominal or pelvic surgery were readmitted a mean of 2.1 times over 10 years for a disorder directly or possibly related to adhesions, and 22.1% of all readmissions occurred in the first year after initial surgery [7,8].

Depending on the cause and location of the adhesions, they can be considered to be either inherent to, and beneficial for, tissue healing, or harmful, through creating complications. Pelvic adhesions are responsible for 15–20% of female infertility cases [5,9], with one study suggesting they cause up to 40% of female infertility cases [10], 80% of chronic postoperative abdominal pain cases, 60% of intestinal obstructions, and can have additional sequelae, such as a reduced range of joint motion [4,10]. Furthermore, they may increase the technical difficulty of subsequent abdominal or pelvic surgery [11]. The effects of adhesions on female infertility are discussed in more detail below.

### Financial Impact of Postoperative Adhesions

Enormous financial consequences of adhesions have been estimated. A study by Ray et al. showed that hospitalizations for adhesiolysis alone during 1988 in the USA accounted for an estimated US $1.18 billion in healthcare expenses. Of this sum, US $925 million went towards hospital costs and US $255 million went towards surgeon fees [8,12]. Subsequent studies have shown that these costs have continued to increase. One follow-up study [13] showed that adhesiolysis expenditures had already reached US $1.3 billion in 1994, and by 2005, these costs had surpassed US $2.3 billion [14]. Moreover, these estimates did not include outpatient costs or indirect costs, such as lost productivity or infertility treatments.

The abovementioned SCAR group also assessed the workload and cost of adhesive intestinal obstruction for two UK hospitals between 1996 and 1997, comparing surgical treatment with conservative care. Of the 110 admissions with adhesive obstruction, 41 (37.3%) were treated surgically, with a mean total treatment cost per admission of £4677.41 and 69 patients (62.7%), with a mean cost of £1606.15 per admission for conservative treatments [15]. Considering inflation, the amounts mentioned have almost doubled in the meantime. [16] The mean length of stay for these admissions was 16.3 days for surgical treatments and 7.0 days for conservative treatments, adversely affecting the length of stay and bed availability [15].

Adhesion-related complications also contribute significantly to increases in healthcare expenses. The direct hospital expenditures in 2005 for adhesive small bowel obstruction alone in the United States were US $3.45 billion, rising to US $5 billion when other pelvic and peritoneal adhesive diagnoses were included [17].

## 5. Causes of Adhesion Formation

In normal healing, tissue injury initiates an inflammatory response and tissue repair starts within minutes, with completion in 3–5 days. Disruption to this process or a mismatch between fibrin deposition and its degradation is believed to cause adhesion formation [18,19,20].

Adhesions are fibrous tissues formed as a result of the internal healing process and inflammatory responses, promoting procoagulatory and antifibrinolytic reactions, and a subsequent significant increase in fibrin formation. They contribute to the body’s defense mechanisms against the causes of inflammation (physical, chemical, infections, etc.) [4,19]. Notably, the formation of these fibrous tissues, which develop inside, and between, organs and structures after inflammation or trauma, is followed by different phases, similar to the normal wound-healing process. They are initially friable and edematous. The inflammation causes an exudation of fibrinogen, which cleaves into fibrin. Fibrin then binds with fibronectin to produce a temporary wound bed. The fibrous tissue shrinks and gains strength as it matures, progressing to mature fibrous tissue in about six weeks [4]. Notably, the pathophysiological mechanisms of adhesion formation have recently been reviewed in further detail, by Goldberg et al. [21].

### 5.1. Abdominal Surgery as a Risk Factor for Adhesion Formation

Abdominal adhesions can occur as a result of any operation or inflammatory cause, including trauma or bleeding. Surgery, particularly open surgery, is the most prevalent recognized cause of adhesions [4]. This includes common procedures, such as appendectomy, cholecystectomy, gastrectomy, hysterectomy, and abdominal vascular operations [5]. Additionally, frequent gynecologic and obstetric procedures, such as myomectomy, tuboplasty, salpingectomy, oophorectomy, and cesarean sections performed through a suprapubic transverse or midline vertical incision, are specifically causative and lead to intraabdominal and pelvic adhesion formation [22]. Myomectomy is particularly adhesiogenic, with 22.6–37.9% of women developing postoperative adhesions, that have multiple contributing factors [23,24].

Postsurgical adhesions can result from incision, cauterization, suturing, and other forms of trauma, where injured tissue surfaces fuse together to form scar tissue. Known factors that have an influence on these adhesions’ formation are the complexity of the operation, the extent of peritoneal trauma, a patient’s poor nutritional status, and the presence of comorbidities in the patient, such as diabetes. Additional factors that may further contribute to postoperative adhesion formation include excessive tension with suturing, where excessive pressure on the sutured region produces ischemia, disrupts lymphatic drainage, and hinders revascularization. Additionally, exposure to foreign bodies such as talc and powders from gloves, lint from abdominal packs, or fibers from disposable paper items may cause adhesions. Such adhesions often contain multiple foreign body granulomas, which suggests a relationship between foreign material, foreign body granulomas, and adhesion formation [5]. Suture granulomas are particularly common in patients who have recently undergone surgery, which further supports the belief that the intra-abdominal presence of foreign material is an important cause of adhesion formation [9]. There are multiple additional risk factors that are more specific to laparoscopic surgery, such as dehydration of the peritoneum through the coupling of dry gas and high insufflation pressure, as well as mesothelial hypoxia through using CO_2_. Conversely, laparotomies carry a risk for dehydration due to the use of light and heat, as well as mesothelial dehydration and abrasion from the use of dry abdominal drapes [3,5].

Although peritoneal adhesions can occur after every abdominal operation, the density, time interval to develop symptoms, and clinical presentations are highly variable and when coupled with additional factors, such as genetic predisposition, this makes their occurrence challenging to predict [25]. Accordingly, factors, such as the type and location of adhesions, as well as the timing and recurrence of adhesive obstructions, remain unpredictable and poorly understood. However, minimally invasive surgical techniques have been shown to reduce adhesion-associated morbidity and mortality [25]. In addition, the local application of a hyaluronic acid barriers has been reported to reduce intraabdominal adhesion formation after laparoscopy [26,27,28] (see below). However, despite recent advances in surgical techniques, there is no consistently reliable strategy that is used to manage postoperative adhesions [25].

### 5.2. Individual Predisposition and Genetic Factors

In addition to procedure-related factors for adhesion development, individual patient-specific factors are assumed. Based on epidemiological data, it appears that the likelihood and severity of adhesions after intra-abdominal surgery varies greatly among patients. Though some individuals develop dense adhesions after surgery, others develop few to no adhesions after similar procedures, by the same surgeon, using comparable surgical techniques. Rather, it seems that adhesions tend to recur in the same patients. These findings are consistent with physiologic and genetic predispositions to the development of postoperative adhesions [29,30].

Unfortunately, mutations increasing the risk for adhesion formation have not been identified in humans. Murine studies have shown that knockout mutations in certain genes (e.g., the Thbs2 gene [31]) can increase the risk of adhesiogenesis. Although screening has not yet identified humans affected by similar mutations, in the future, these genes and their associated pathways may be used for counseling, as predictive genomics advances as a field [29].

Unfortunately, even if prediction becomes more accurate, therapy may be difficult, owing to the interplay between the numerous growth factors and cytokines that interact within the adhesion cascade. However, genetic insights may help in targeting those who should receive barrier therapies, as well as in designing more effective trials, by excluding low-risk patients who would bias outcomes towards the null [29].

### 5.3. Other Causes of Adhesions

While sexually transmitted diseases (STDs), such as chlamydia, gonorrhea, and trichomoniasis, are associated with tubal adhesions and occlusions [32], the nature and nuances of infectious etiologies are a better fit for a separate article.

With an incidence of almost 40% [33], endometriosis is among the most frequent causes of intra-abdominal adhesions. The prevalence of endometriosis in women with infertility is even estimated to be up to 50% [34]. However, since the focus of this review is not on endometriosis-associated adhesions, we would also like to refer to further literature on this topic.

## 6. Diagnostics

One of the challenges in assessing the true incidence of adhesiogenesis is that adhesion formation can be asymptomatic or, alternatively, the symptoms are vague and mild enough to not warrant investigation. It is only in the more pronounced cases, leading to surgical assessment, where symptoms can manifest, such as through chronic pelvic pain, small bowel obstruction, or infertility [35,36].

Laparoscopy is considered the gold standard procedure for diagnosing abdominal and pelvic adhesions, with easier assessment than laparotomy for the whole abdomen through distention and magnification, as well as reduced pain through smaller incisions [37,38,39,40]. The main downside to laparoscopy is that it is invasive, with patients being exposed to general anesthesia, potential surgical complications, and further adhesion formation [39].

In spite of the limits to other diagnostic tools, clinical tests and imaging procedures, laparoscopy allows for the most definitive and, ultimately, reliable diagnosis [3]. Nevertheless, there have been numerous studies conducted, in order to determine as to whether less invasive methods might also provide useful diagnostic results. Some studies consider transvaginal hydro-laparoscopy, sometimes also called “fertiloscopy”, to be a safer, cheaper, and less invasive diagnostic method for finding pelvic adhesions [41,42,43]. In transvaginal hydro-laparoscopy, access to the Douglas pouch and, thus, to the pelvic cavity, is gained, by inserting a single-needle, dilating trocar system through the posterior fornix. The examination is performed under either general or local anesthesia, and patients can be discharged immediately after the procedure is completed [41]. The low complication rate can be further reduced by a transabdominal ultrasound-guided vaginal access, which is especially useful in patients with a retroverted uterus [44]. A multicenter study on 43 infertile patients undergoing both transvaginal hydro-laparoscopy and standard laparoscopy was able to show that mild ovarian adhesions can be more accurately diagnosed by transvaginal hydro-laparoscopy [45]. The number of subtle ovarian adhesions found during transvaginal hydro-laparoscopy significantly exceeded the number of findings during standard laparoscopy (present in 12 vs. 4; absent in 9 vs. 17). Although this study has shown that transvaginal hydro-laparoscopy enables the better assessment of mild ovarian adhesions than that by standard laparoscopy [43], due to the restricted view, abdominal adhesions cannot be examined using transvaginal hydro-laparoscopy [46].

Another approach is the use of radiologic diagnostic methods. In a prospective study, 60 female patients were examined for abdominal wall adhesions before undergoing laparoscopic surgery. The presence of the visceral sliding sign (where intraabdominal organs freely move beneath the abdominal wall during respiration or manual compression, and where a lack of mobility is meaningfully associated with adhesions) was measured using transabdominal ultrasonography (TAU) and magnetic resonance imaging (cine MRI). After laparoscopy, the results were then compared again. TAU showed an accuracy of 81.3% and cine MRI was 72.4% accurate, but with sensitivities of only 24% and 21.5%, respectively, when compared to the laparoscopic findings. Due to the low sensitivity, neither radiologic method was considered sufficient for the diagnosis of adhesions. Nevertheless, both non-invasive methods have shown a reasonable specificity in detecting adhesion-free areas [47].

The visceral sliding sign is an important sonographic indicator of adhesions. A recent meta-analysis of 21 studies concluded that the visceral slide assessment of the periumbilical area with ultrasound had a high negative predictive value for the absence of periumbilical bowel adhesions in patients at risk for adhesions. In detail, ultrasound assessment for periumbilical bowel adhesions had a combined sensitivity of 95.9% (95% confidence interval, 82.7–99.1%) and specificity of 93.1% (85.1–96.9%) [48]. Because the visceral sliding sign is better for assessing some pelvic regions than others, it has been suggested that the sliding sign should be an adjunct to surgical assessment, for the most thorough approach to evaluation. Preoperative ultrasound may help reduce complications and determine the optimal site for placing the first trocar in laparoscopy [5,48,49,50].

Fewer studies have been performed for the sliding sign in transvaginal ultrasounds for predicting pelvic adhesions. However, a large prospective, multicenter, double-blind study revealed that the sliding sign had a sensitivity of 96.3% and a specificity of 92.6% in predicting pelvic adhesions, with a significant relationship between adhesions in each compartment and the sliding sign [51].

Another recent study with similar methods yielded comparable results. In this investigation, 131 women suffering from endometriosis underwent transvaginal ultrasonography with mapping, in order to determine the presence or absence of adhesions, by using the sliding sign technique and a scoring system (0–10) preoperatively. Afterwards, the ultrasound findings were compared with the surgical findings, yielding a sensitivity and a specificity to adhesion mapping of 80.4% and 86.1%, respectively. This correlates with adhesion mapping having an accuracy of 83.9% relative to surgery [52].

However, despite these methodical progresses, laparoscopy remains the gold standard procedure for the diagnosis of intra-abdominal adhesions [3].

## 7. Effects of Abdominal Adhesions on Infertility

Adhesions can limit the movement and function of organs, ligaments, muscles, and other anatomical structures. This can distort the pelvic anatomy and restrict blood supply to pelvic tissues, hindering conception, even with in vitro fertilization [53,54]. Infertility-associated adhesions may form on uterine walls and ligaments or within the cervix, hindering the progression of sperm to the uterus and Fallopian tubes, as well as potentially increasing uterine spasms, implantation problems, and miscarriage, as well as otherwise hindering conception. Paraovarian adhesions may limit the ability for the fimbria to pick up the oocyte [3,53]. When they occur at the distal part of the fallopian tube, they restrict the tentacle-like grasping of the ovum by the fimbria, increasing its risk of being wasted in the abdominal cavity. However, if they occur on the inner or outer side of the Fallopian tube, they can lead to partial or total tubal occlusion, decreasing the probability of conception, while increasing the risk of ectopic pregnancy. Adhesions may also hinder ovarian access for oocyte aspiration [53,55].

## 8. Adhesion Prevention and Therapy

There are several approaches used in order to minimize adhesion formation. The clearest is avoiding unnecessary surgical procedures [56]. Other strategies include certain minimally invasive surgical techniques, adhesion barriers, or other treatments, in order to suppress inflammation, manipulate coagulation, and increase fibrinolytic activity [19,36].

### 8.1. Avoiding Unnecessary Interventions

A prospective study has been able to show that the incidence of adhesions increases with the number of previous operations. At the time of their first laparotomy, 10.4% (12 of 115) of the patients studied already had adhesions, typically due to previous inflammatory processes. Of the 210 patients with prior laparotomy, the incidence of adhesions was 93% (195 of 210), with the incidence already exceeding 91% in the 150 patients with only one prior laparotomy (137 of 150) [6].

### 8.2. Surgical Techniques

In recent times, a variety of products has been introduced by the medical industry, in order to prevent adhesions. These new instruments have led surgeons to pay less attention to the most important and classic rules for adhesion prevention. However, good surgical technique should continue to be promoted as a means of preventing adhesions. This includes keeping direct tissue trauma to a minimum, avoiding desiccation, achieving optimal hemostasis, and minimizing the risk of infection [18]. Laparoscopy reduces postoperative adhesions relative to laparotomy [21].

Several factors increasing the likelihood of intraperitoneal adhesion formation have been identified over the last decades. These include, among others, peritoneal injury, the approximation of two injured serosal surfaces, and the detrimental effects of ischemia. It was also found that clotted blood adheres firmly to a desiccated serosal surface, whereas heparinized blood does not cause adhesion [57,58,59,60]. Based on these findings, ‘‘microsurgery’’, a set of principles primarily focused on the prevention of postoperative adhesions, was developed, in order to improve the outcomes of fertility surgery. These principles include gentle tissue handling, meticulous hemostasis that minimizes adjacent tissue damage, and the avoidance of foreign body contamination by using talcum-free gloves and lint-free surgical pads [60]. Another major component of these principles is the frequent intraoperative irrigation with heparinized fluid, in order to prevent tissue desiccation and blood clots, a practice that is not routinely performed in gynecologic surgery [18,60]. Additionally critical is the complete excision of abnormal tissues. Depending on the circumstances, the precise alignment and approximation of tissue planes, as well as the concept of keeping denuded areas separated by temporary adnexal or ovarian suspensions, are further essential parts of these tenets [60]. Another suggested measure is the postoperative administration of dexamethasone [60,61,62,63], although the effectiveness of this procedure is partly controversial. [11] All these measures are cornerstones of reproductive microsurgery, whether performed by means of open access or laparoscopy [60]. The introduction of these microsurgical principals in gynecological surgery in the 1980s led to a re-evaluation of the concepts and rules of surgical principles, in order to avoid adhesions. By following the mentioned measures, an actual reduction of the adhesion rate compared to conventional surgery, as well as a subsequent improvement of fertility rates, can be achieved, as studies have shown [60,64,65]. Unfortunately, after this period, there were insufficient articles to refresh this important concept in the field of gynecological surgery.

It is undeniable that the standard of care for most surgeries has shifted from laparotomy to laparoscopy. Despite widespread expectations that this increase would result in a significant reduction in postoperative adhesions, studies have shown conflicting results [18]. Some showed a small but significant improvement in the incidence of adhesions and pregnancy rates with laparoscopy [10,66,67], while others showed no difference in morbidity, and comparable risks of adhesion-related readmissions with laparoscopic surgery relative to open surgery [18,68], despite the patients included in these studies undergoing similar procedures. Both general surgical procedures, such as appendectomies and cholecystectomies, and gynecological procedures, including myomectomies, salpingectomies, Cesarean sections, and others, were included in these analyses [10,66,67,68].

In their review, Kavic and Kavic summarized 12 studies published between 1989 and 2000, evaluating the impact of laparoscopy versus laparotomy on adhesions. The heterogeneity of the study design, assessment end points, and the use of animal models left the authors unable to perform a meta-analysis; yet, the majority of studies (58.3%; 7 of 12) found laparoscopy to be beneficial in reducing adhesions, including in three clinical studies on humans. However, four studies did not show any difference between laparoscopy and laparotomy, and one study demonstrated laparotomy to be less adhesiogenic than laparoscopy [66]. Another review on the same topic by Gutt et al. identified 15 studies over a similar period (1987 to 2001). Of these 15 studies, 11 were, in fact, the same as in the review by Kavic and Kavic. Notably, 60% of the examined studies (9/15) found fewer adhesions following laparoscopy than following laparotomy, while other five studies had revealed similar adhesion rates for the two surgical methods. Four of these five experimental studies used a standardized method of tissue injury, laparotomically and laparoscopically. This defeated the main advantage of laparoscopy, as it is thought to reduce adhesions by minimizing the injury to adjacent tissue, which explains the similar results. The fifth study was performed under special conditions of peritonitis, where the surgical intervention became insignificant when compared to the septic injury of the whole peritoneum [10].

A systematic review and meta-analysis by Broek et al. identified 27 studies about different surgical techniques used to reduce adhesion formation, including six articles, which compared adhesion formation between laparoscopy and laparotomy. However, none of the compared techniques reduced adhesion-related infertility, although the incidence of adhesions was lower after laparoscopy when compared to open surgery (relative risk 0.14; 95% confidence interval: 0.03–0.61; *p* = 0.008). Moreover, the meta-analysis provides little evidence to support the surgical principle that the use of less-invasive techniques, inducing less tissue damage, reduces the extent and severity of adhesions. Possible reasons for this are that the extent of serosal wound surfaces is often comparable in open and laparoscopic procedures, and that the CO_2_ pneumoperitoneum may even injure the entire peritoneal surface and promote adhesion formation at distant sites [67]. Additionally, as mentioned above, the CO_2_ pneumoperitoneum, the higher intra-abdominal pressure, and the laparoscope light are associated with peritoneal ischemia, decreased fibrinolysis, and increased adhesion formation [3,5,8,67,69]. However, studies have shown that using warmed, humidified insufflation gas may reduce adhesions [8,18]. Other studies have shown that the combined application of dexamethasone, cooling the peritoneal cavity with cold saline irrigation, and adding NO_2_ or O_2_ to the CO_2_ pneumoperitoneum can lead to a reduction in adhesions by 85% in mouse models [18,63].

### 8.3. Adhesion Barriers and Pharmacological Agents

Adhesion barriers aim to prevent healing tissues from touching one another during the 3 day to 5 day period of peritoneal remesothelialization. They can be solid, fluid, or gel [18]. Fluid and gel agents have a similar mechanism of action as barrier agents. Pharmacological agents, however, aim to modify aspects of the healing process, so as to reduce adhesion formation. Although many studies have demonstrated no serious adverse effects associated with these barriers, there is still insufficient evidence regarding their effectiveness [18,28,70].

A recent review by the Cochrane Gynecology and Fertility Group on barrier agents for adhesion prevention summarized 19 randomized controlled trials, comparing the different types of agents currently in use. A total of 1316 women undergoing gynecologic surgery participated in these studies. In all included trials, barrier agents were compared either with each other, or with no treatment. However, the relative effectiveness of the agents was only significant in the interventions where they were compared with each other, rather than with the placebo, and all the evidence was of very-low to moderate quality; thus, the results should be viewed with caution. There is evidence that a collagen membrane containing polyethylene glycol plus glycerol (Gynecare Interceed^®^, Ethicon, Johnson & Johnson Surgical Technologies, Bridgewater, NJ, USA) may be more effective than no treatment in reducing adhesion formation after pelvic surgery. Other low-quality evidence suggests that oxidized regenerated cellulose (Surgicel^®^, Ethicon, Johnson & Johnson Surgical Technologies, Bridgewater, NJ, USA) may reduce the incidence of new adhesion formation when compared with no treatment during laparotomy. Because of insufficient evidence, it is not possible to draw conclusions about the relative effectiveness of these interventions. The most frequent limitations were imprecision, such as few participants and wide confidence intervals, as well as insufficient detail in the study methods. In addition, most trials had been commercially funded (13/19; 12 of which were sponsored by companies that manufactured the adhesion agents). Thus, publication bias could not be excluded. Notably, no adverse events directly related to the adhesion agents were reported [28].

Along with the first review, the Cochrane Gynaecology and Fertility Group also published a review on fluid and pharmacological agents for adhesion prevention, including 32 randomized controlled trials, with a total number of 3492 women. The results of 23 studies (2796 women) were pooled, whereas the results of the remaining nine trials could not be summarized because of insufficient published information. The quality of the evidence ranged from very-low to high. Similar to the other review, the main reason for poor evidence was imprecision, with small sample sizes and wide confidence intervals that crossed the line of no effect, as well as the insufficient reporting of methods. However, four studies of high-quality evidence identified effective fluids, and another five high-quality studies noted that gels appeared to be effective in reducing the incidence of adhesions during second-look laparoscopy when compared with no treatment. This suggests that in women with an 84% chance of forming adhesions during second-look laparoscopies, with no treatment, the use of fluid agents would result in this likelihood decreasing to a 54% to 75% chance of forming adhesions, and the use of gel agents would reduce the probability to a 39% to 75% chance of forming adhesions. Once again, no adverse events directly related to the agents were reported. However, it remains uncertain as to whether fluid agents, gel agents, or steroids affect clinical pregnancy rates compared with no treatment. More information is needed to determine as to whether they affect pelvic pain or live birth rates. Therefore, large, high-quality studies should be conducted, in which investigators use the standardized method for measuring adhesions (the modified AFS score) [70].

However, a study by Tulandi et al. showed contrary results [71]. In this study, the use of an adhesion barrier was associated with a slightly higher risk of increased postoperative fever, as well as ileus or bowl obstruction, especially after myomectomy or hysterectomy by laparotomy. However, the results of this study should be taken with caution because of their considerable limitations; it was conducted retrospectively, without detailed information about the operations performed, as well as having unclear diagnostic criteria for ileus or bowel obstruction and, most importantly, a lack of knowledge about the types of adhesion barriers used [71].

Despite these mainly favorable findings, adhesion barriers continue to be used infrequently, either because surgeons are unaware of the magnitude of the burden caused by adhesions or because they do not believe in the efficacy of adhesion barriers [18]. A Dutch study surveyed 380 surgeons on the topic of postoperative adhesions. The study showed that although a large number of surgeons perceive postoperative adhesions as a complication (88.1%), only about one-third (38.8%) have a sound knowledge of their complications. One-third of surgeons (32.5%) do not even inform their patients preoperatively about adhesions and their consequences, and about another third (29.8%) do not believe in the efficacy of anti-adhesive products [72]. It has been argued that the negative effects of adhesions are so great that adhesion barriers should be used routinely to reduce morbidity and costs, even if the effects may be modest. Ten Broek et al. estimated, based on costs in the United Kingdom and the Netherlands, that adhesion barriers could decrease the direct hospital costs of adhesion-related complications, within the first 5 years after surgery, by $328–680 for open surgery and by $63–82 for laparoscopic surgery [18,73].

## 9. Treatment Options

### 9.1. Adhesiolysis

The primary therapy for adhesions is surgical adhesiolysis. In this regard, adhesions can be removed using various techniques, such as sharp dissection, electrosurgery, and different types of lasers. The method used to remove the adhesions does not play a significant role, as an animal study showed no differences in the effectiveness of electrocautery, a CO_2_ laser, and an Nd:YAG laser, although it should be noted that the Nd:YAG laser surgery was slower and caused more tissue damage [74,75]. However, the timing of adhesiolysis and, consequently, the consistency of the adhesions seems to make a significant difference in terms of the likelihood of adhesion reformation. Thin and filmy adhesions, seen in laparoscopies performed within a couple of days or weeks after initial surgery, are a lot less likely to reform after lysis than denser and more vascular adhesions that can be seen with later second-look laparoscopies (e.g., reformation in 8 out of 9 patients with dense/thick adhesions (88.9%) vs. 5 out of 22 patients with thin/filmy adhesions (22.7%) [76]) [76,77,78].

Regarding fertility, successful pregnancy outcomes after adhesiolysis are controversial. While one study has shown that the short-interval lysis of mild, filmy adhesions within 3–4 weeks after surgery seems to show no benefit on pregnancy rates compared to expectant management (pregnancy rates over 6 months: 47% (9/19) in the adhesiolysis group vs. 55% (11/20) in the expectant-management group; *p* > 0.05) [79], other studies [80,81] have shown a positive effect of adhesiolysis on pregnancy rates. A retrospective study in 1990 evaluated the efficacy of adhesiolysis in infertile women. Of 147 infertile women found to have periadnexal adhesions at laparoscopy and who otherwise had unexplained infertility, 69 (47%) were treated by laparotomy and salpingo-ovariolysis, and 78 (53%) were not treated, with no significant difference between the degree of adhesions in the two groups. Pregnancy rates in the adhesiolysis-group were 32% at 12 months and 45% at 24 months, compared with 11% at 12 months and 16% at 24 months in the women who were left untreated (*p* < 10^–6^) [11,81].

The main problem with adhesiolysis is that the intervention is also a surgical procedure that can subsequently lead to further adhesions. For this purpose, a review compared 12 studies in which laparoscopic adhesiolysis was performed, mostly in human clinical trials. In these studies, new adhesion formation after laparoscopic adhesiolysis was identified in 20% to 97% of patients [64]. Another issue and feared complication of adhesiolysis is unintentional enterotomy. In a retrospective study, 19% of reoperations (52 of 270) resulted in an inadvertent enterotomy [82]. In particular, laparoscopic adhesiolysis is associated with a significant risk of bowel perforation. It can occur during the establishment of the pneumoperitoneum or during adhesiolysis itself. Diathermic lesions of the bowel are of particular concern, because perforation does not occur immediately [74]. However, in vitro fertilization can produce similar pregnancy rates without the aforementioned risks of surgery [66,83]. An additional consideration is that studies publishing outcomes after second-look adhesiolysis often involve expert hands, relative to the typical quality of technique used. Additionally, another factor further limiting the expected advantages of surgery is that for patients with previous pelvic inflammatory disease, peritubal adhesiolysis will not correct intraluminal ciliary trauma. Accordingly, adhesiolysis has the greatest value for neosalpingostomy, less for peritubal adhesions not involving the fimbriae, and is least able to restore normal intraluminal function.

### 9.2. Other Discussed Forms of Therapy

The approach of physical therapy to resolve adhesions in infertile women has also been investigated. In his study, Wurn was able to show that mechanical treatment, so-called “site-specific manual soft-tissue therapy”, led to a higher pregnancy rate in infertile women (10 of 14; 71.4%) and IVF patients (clinical pregnancies in 22 of 33 embryo transfers) compared to other techniques. The aim of the intervention was to create the microfailure of collagenous cross-links, the “building blocks” of adhesions in the pelvic region and around the fallopian tubes. Nevertheless, the study quality is rather poor, as a large proportion of the treated patients (51.3%) did not have adhesions confirmed by laparoscopy (gold standard), but only had a very high probability (due to previous surgery, infectious or inflammatory disease, etc.) of intra-abdominal adhesions. Only 48.7% of the treated patients had a confirmed diagnosis of adhesions, whereby the paper also does not mention by which diagnostic procedure it was made. Another fundamental weakness of the work is that the likely affected adhesion sites were determined solely by history and not by any imaging techniques. However, the great advantage of this method is that it is a nonsurgical, noninvasive manual technique, with no risks, and few, if any, adverse side effects or complications [53]. With a small sample size for both spontaneous pregnancy and IVF, further studies need to be done, particularly when the breadth of efficacy to many clinicians seems to exceed biologic plausibility.

Ozone is currently being evaluated as a new complementary therapeutic agent for female infertility. A review of the current literature on the effect of ozone therapy on various factors that could potentially affect female fertility, has revealed that among other outcomes, ozone therapy might lead to a lower incidence of the formation of pelvic adhesions [84].

Oxidative stress, which occurs during laparotomy and laparoscopy, is thought to play an important role in adhesion formation [69]. Ozone therapy aims to reduce inflammation and increase the activity of antioxidant enzymes, such as glutathione peroxidase, by altering the production of reactive oxygen species [84].

A randomized controlled trial was conducted, in order to evaluate the efficacy of ozone therapy in a rat model with experimental uterine adhesions, induced by bipolar coagulation [83]. The ozone therapy was then given to one group intraperitoneally at 0.7 mg/kg daily as a single dose for three days, followed by the measurement of various parameters of oxidative stress, including TNF-α. The results showed that the group receiving ozone therapy had a significantly lower macroscopic adhesion score and lower peritoneal TNF-α levels compared to the control group (*p* < 0.001), which indicates that there may be potential for ozone therapy to reduce postoperative uterine adhesions by modulating TNF-α levels and by altering the oxidative state [84,85].

Ultimately, most of the data regarding ozone therapy were collected in animals, and very few human studies are found in the literature. Accordingly, there is a need for human studies on the effects of ozone therapy on female fertility and pelvic adhesions [84].

## 10. Conclusions

Postoperative adhesions are a troublesome side effect of surgery, occurring in up to 97% of cases after open gynecologic procedures. However, they are certainly very important to consider after any kind of surgery. Besides small bowel obstruction and chronic pelvic pain, they may account for up to 40% of cases of female infertility. Moreover, adhesion-related complications account for a meaningful source of hospital readmissions, adding to both postoperative morbidity and healthcare costs. This makes postoperative adhesions a burden not only for the affected patients but also for the healthcare system. Laparoscopy remains the gold standard for diagnosing intra-abdominal adhesions, although it is an invasive procedure, and intervention can lead to further adhesion formation. Transvaginal hydro-laparoscopy has limitations, but may be a safer, cheaper, and less-invasive diagnostic method. Radiographic techniques have a role in diagnosing adhesions, but pose greater risks for false-positives and -negatives, even if they have lower costs and of a lower risk than surgical assessment. The intraoperative application of barrier agents, especially gel barriers, has also been shown to be a beneficial way to reduce postoperative adhesions. Ultimately, as a field, we may need to invest as much research as possible in the consequences of our surgery, as in the potential initial need for it, as well as if we are to find an appropriate balance of success and safety for our patients.

## Data Availability

All the data is available within the study. This process can be initiated upon request to the corresponding author.

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
