# Peer review of "An Overview of Postoperative Intraabdominal Adhesions and Their Role on Female Infertility: A Narrative Review"

_jcm, 2023, doi:10.3390/jcm12062263_

Round 1

Reviewer 1 Report

Very detailed review regarding adhesions. Surgical adhesions and the rules to decrease the incidence are almost forgotten. This review is excellent to refresh the memory for the basic learning of adhesion prevention and treatment.

1.Pelvic or abdominal adhesions are very important to  consider after any kind of surgery.

2.In late 1980’s the introduction of microsurgery in the gynecological surgery leaded to review the main concepts and rules of surgical principles to avoid adhesions.After this period there are inadequate articles to refresh this important concept in the field of gyn surgery.

3. Recently there are many add-backs to avoid adhesions introduced by the industry. This new instruments caused surgeons to pay less attention to main and classical rules for adhesion prevention.

4.The authors underlining the importantance of surgical principles and rules to avoid adhesions are still much more important than some add-backs.

5.Conclusions are consistent with evidence and arguments.

6.References are good enough

7. This review is important to compare the add-backs and classical principles to avoid adhesions. Adhesions may cause several problems  after surgery that may down credit the the success of surgery. So every surgeon should obey the rules of clean surgery during the operation.

Author Response

Very detailed review regarding adhesions. Surgical adhesions and the rules to decrease the incidence are almost forgotten. This review is excellent to refresh the memory for the basic learning of adhesion prevention and treatment.

- We thank the reviewer for the appreciative overall assessment of our manuscript.

1.Pelvic or abdominal adhesions are very important to  consider after any kind of surgery.

- We have underlined this point once again in the Conclusion: "Postoperative adhesions are a troublesome side effect of surgery occurring in up to 97% of cases after open gynecologic procedures. However, they are certainly very important to consider after any kind of surgery."

2.In late 1980’s the introduction of microsurgery in the gynecological surgery leaded to review the main concepts and rules of surgical principles to avoid adhesions.After this period there are inadequate articles to refresh this important concept in the field of gyn surgery.

3.Recently there are many add-backs to avoid adhesions introduced by the industry. This new instruments caused surgeons to pay less attention to main and classical rules for adhesion prevention.

- We thank the reviewer for this statements and take the liberty of using his/her wording. We have added these comments to point 8.2. Surgical techniques.

4.The authors underlining the importantance of surgical principles and rules to avoid adhesions are still much more important than some add-backs.

5.Conclusions are consistent with evidence and arguments.

6.References are good enough

7.This review is important to compare the add-backs and classical principles to avoid adhesions. Adhesions may cause several problems after surgery that may down credit the the success of surgery. So every surgeon should obey the rules of clean surgery during the operation.

- We would like to thank the reviewer again for the appreciation and affirming comments.

Reviewer 2 Report

The authors have done an excellent job reviewing the literature on the role of post-operative adhesions.

It is obvious however, that the scope of this article does not center on fertility. There, the title should be changed.

Moreover, I believe the search terms should also include endometriosis and hysterectomy; not vulvectomy. Literature related to these entities should be included. For example, section 5.1 does not even mention endometriosis.

Lastly in regards to English Language and Style, I would change the use of "great deal of money" in the abstract.

Author Response

The authors have done an excellent job reviewing the literature on the role of post-operative adhesions.

- We thank the reviewer for the appreciative overall assessment of our manuscript.

It is obvious however, that the scope of this article does not center on fertility. There, the title should be changed.

- This is a good point and we have changed the title accordingly: "An overview of postoperative intraabdominal adhesions and their role on female infertility: a narrative review". However, we want to keep the aspect of infertility in the title, in line with the special issue. We ask for the reviewer's kind understanding.

Moreover, I believe the search terms should also include endometriosis and hysterectomy; not vulvectomy. Literature related to these entities should be included. For example, section 5.1 does not even mention endometriosis.

- We thank the reviewer for the helpful suggestion. We changed the search term from vulvectomy to hysterectomy as proposed. Using this search query, we were shown an additional 9 articles, which unfortunately did not turn out to be suitable for our review.

- With regard to endometriosis, due to the large extent of this topic, we had decided against dealing with it, because it would go beyond the scope of this review. We have added a short paragraph to point 5.3. Other causes of adhesions: "With an incidence of almost 40% [84], endometriosis is among the most frequent causes of intra-abdominal adhesions. The prevalence of endometriosis in women with infertility is even estimated to be up to 50% [85]. However, as the focus of this review is not on endometriosis-associated adhesions, we would also like to refer to further literature on this topic."

 We also kindly ask for the reviewer's indulgence at this point. 

Lastly in regards to English Language and Style, I would change the use of "great deal of money" in the abstract.

- Thank you for the advice, we have changed this expression to a more suitable one: "considerable amount of money".
